# Metagenomic Insights Reveal the Microbial Diversity and Associated Algal-Polysaccharide-Degrading Enzymes on the Surface of Red Algae among Remote Regions

**DOI:** 10.3390/ijms241311019

**Published:** 2023-07-03

**Authors:** Xiaoqian Gu, Zhe Cao, Luying Zhao, Dewi Seswita-Zilda, Qian Zhang, Liping Fu, Jiang Li

**Affiliations:** 1Key Lab of Ecological Environment Science and Technology, First Institute of Oceanography, Ministry of Natural Resources, Qingdao 266061, China; guxiaoqian5843@163.com (X.G.); caozhe_071@163.com (Z.C.); zhaoluying0621@163.com (L.Z.); zhangqian_72@163.com (Q.Z.); liping_fu@163.com (L.F.); 2CAS and Shandong Province Key Laboratory of Experimental Marine Biology, Center for Ocean Mega-Science, Institute of Oceanology, Chinese Academy of Sciences, Qingdao 266071, China; 3Research Center for Deep Sea, Earth Sciences and Maritime Research Organization, National Research and Innovation Agency (BRIN), Jl. Pasir Putih Raya, Pademangan, Jakarta 14430, Indonesia; dewi052@brin.go.id

**Keywords:** metagenomics, red algae, microbial diversity, algal-polysaccharide-degrading enzymes

## Abstract

Macroalgae and macroalgae-associated bacteria together constitute the most efficient metabolic cycling system in the ocean. Their interactions, especially the responses of macroalgae-associated bacteria communities to algae in different geographical locations, are mostly unknown. In this study, metagenomics was used to analyze the microbial diversity and associated algal-polysaccharide-degrading enzymes on the surface of red algae among three remote regions. There were significant differences in the macroalgae-associated bacteria community composition and diversity among the different regions. At the phylum level, Proteobacteria, Bacteroidetes, and Actinobacteria had a significantly high relative abundance among the regions. From the perspective of species diversity, samples from China had the highest macroalgae-associated bacteria diversity, followed by those from Antarctica and Indonesia. In addition, in the functional prediction of the bacterial community, genes associated with amino acid metabolism, carbohydrate metabolism, energy metabolism, metabolism of cofactors and vitamins, and membrane transport had a high relative abundance. Canonical correspondence analysis and redundancy analysis of environmental factors showed that, without considering algae species and composition, pH and temperature were the main environmental factors affecting bacterial community structure. Furthermore, there were significant differences in algal-polysaccharide-degrading enzymes among the regions. Samples from China and Antarctica had high abundances of algal-polysaccharide-degrading enzymes, while those from Indonesia had extremely low abundances. The environmental differences between these three regions may impose a strong geographic differentiation regarding the biodiversity of algal microbiomes and their expressed enzyme genes. This work expands our knowledge of algal microbial ecology, and contributes to an in-depth study of their metabolic characteristics, ecological functions, and applications.

## 1. Introduction

Marine algae produce approximately half of the global primary productivity, and the algae polysaccharides produced by these algae, together with polysaccharide-degrading heterotrophic microorganisms and their polysaccharide-degrading enzymes, form the largest and most dynamic carbon cycle on Earth [1,2]. However, little is known about this microbial-driven system of marine carbon metabolism.

Marine microorganisms play important roles throughout all marine ecological processes; therefore, an increasing amount of attention is being paid to the study of their populations and functions [3,4,5]. However, the microbial communities on the surface of algae are still not fully understood, despite their high biodiversity; furthermore, they are markedly different from those that live freely in seawater [3]. Microorganisms on the surface of algae, through a series of complex interactions with their hosts, constitute a rich source of bioactive compounds and specific polysaccharide active enzymes [6]. It is generally believed that the secondary metabolites of microorganisms depend on a range of available organic carbon sources produced by the host algae [7]. Red macroalgae are a source of nutrition for many marine microorganisms. Carrageenan and agar are types of sulphated galactose, which are the main cell wall components of red algae [8]. The study of macroalgae-associated bacterial communities and their relationship with algal hosts provides important information on the ecological function of algae in aquatic ecosystems [8]. In addition, because these microbes interact with algae in a number of complex ways, together they form an important source of novel bioactive compounds such as agarase and carrageenase, with potential applications [6]. Agarose and carrageen, and their oligosaccharides, are widely used in food, medicine, agriculture, cosmetics, and other fields, and have huge application potential and market prospects [9,10].

Algae host a large number of bacteria on their surface (up to 10^6^ cells·cm^−2^), which varies greatly according to species location and climatic conditions [4,6,11,12]. In addition, bacteria can become endosymbionts in algal cells and freely grow in the “phycosphere”. These microorganisms have a very complex relationship with algae [13,14]. Associated bacteria play a crucial role in the normal growth and development of algae [15]. However, some bacteria can also become pathogenic [16]. Due to the different thallus components, macroalgae can also regulate the associated bacteria in various ways, and the community structure of macroalgae-associated bacteria on the surface of macroalgae also differs [17]. In addition, seasonal variations, spatial differences, and environmental factors can also affect the composition of macroalgae-associated bacteria [18,19]. The microbial community composition and diversity of the same algal species from different sea areas can be significantly different. The functional selection of different seaweed species from the same sea area is also closely related to their habitat, and there are large differences in the dominant species and diversity [20,21]. The community composition of macroalgae-associated bacteria is not only related to the environment, but is also affected by the host, which is in accordance with natural selection.

In recent years, an increasing amount of research has investigated macroalgae-associated bacteria and algal-polysaccharide-degrading enzymes [22,23]. Numerous macroalgae-associated marine microorganisms contain undeveloped enzymes such as glycoside hydrolase (GH) and polysaccharide lyase (PL) [24], which are involved in important metabolic pathways, and their characterization has the potential to provide new insights into the marine carbon cycle. However, the existing research mostly focuses on certain regions, lacking a comprehensive understanding of the biodiversity and composition of macroalgae-associated microbiology among remote regions, in part due to limited microbiological sampling of macroalgae, such as in the remote and hostile Southern Ocean. The spatial transformation of microbial communities on macroalgae’s surface may be affected by the complex interactions among abiotic and spatial factors; therefore, it is of great significance to study the relative effects of different driving forces on microbial diversity and community composition on the surfaces of macroalgae across remote and distinct environments.

The application of metagenomics significantly promotes the study of marine microbial diversity and biological function, and greatly increases the opportunities for the discovery of potential new enzymes [25,26]. Metagenomics not only provides more information for understanding the microbial diversity that flourishes in marine systems, but can also further reveal the taxonomic composition and functional potential of microbial communities, and provide genomic information for understanding the structure and biogeochemical characteristics of macroalgae ecosystems [6,27].

In this study, metagenomics was used to analyze the microbial diversity, functional diversity, and associated algal-polysaccharide-degrading enzymes on the surface of red algae among three remote regions, and the effects of spatial distance on microbial diversity and associated algal-polysaccharide-degrading enzymes, especially the effects of environmental factors (pH and temperature, etc.), are discussed. This study effectively extends our understanding of the diversity of algal surface microorganisms and their polysaccharide-degrading enzymes. This work paves a new path for the exploitation and utilization of macroalgae resources in the future and the exploitation of excellent algal-polysaccharide-degrading enzymes from special environmental sources.

## 2. Results and Discussion

### 2.1. Characteristics of the Nine Macroalgae Samples

The physiochemical parameters of the seawater overlying the red algae among the three remote regions differed, with the following ranges (Table 1): water temperature −1.34–27.9 °C, salinity 32.4–34.7‰, pH 8.01–8.79, dissolved oxygen (DO) 8.07–8.79 mg/L, latitude 0.6–62.2°, and longitude 58.95–127.86°. These physicochemical parameters of water samples can serve as an imperfect proxy for the properties of macroalgae samples, due to wave and current distributions.

### 2.2. Sequencing and Metagenomic Assembly

The Illumina HiSeq sequencing platform produced a total of 81,611.46 Mbp of raw data (average data volume was 9067.94 Mbp). After quality control, 81,245.68 Mbp of clean data were obtained (average data volume was 9027.30 Mbp), and the effective data rate of the quality control was 99.55% (Appendix A). After single-sample assembly and mixed assembly, a total of 2,155,261,173 bp scaftigs were obtained with an average length of 1178.16 bp, maximum length of 514,363 bp, N50 length of 1230.89 bp, and N90 length of 573.11 bp. Scaffolds were interrupted at N to generate scaftigs, yielding a total of 2,155,261,173 bp scaftigs with an average length of 1178 bp; 1231 bp for N50 and 573 bp for N90 (Appendix A).

### 2.3. Microbial Diversity and Community Composition

There were 2,485,201 prediction genes after the original redundancy removal, among which the number of ORFs annotated in the NR database was 1,781,793 (71.70%). The proportions were 83.52%, 79.82%, 74.49%, 69.68%, 66.09%, 54.98%, and 44.15% at the kingdom, phylum, class, order, family, genus, and species levels, respectively.

Among all seawater microorganisms, the community structure of macroalgae-associated bacteria has certain specificity, and the two can be bidirectionally selected through interactions [28]. Therefore, studying the macroalgae-associated bacteria community is a necessary prerequisite to understanding the relationship between algae and bacteria. At the phylum level, Proteobacteria (validly published name: Pseudomonadota), Bacteroidetes (validly published name: Bacteroidota), and Actinobacteria (validly published name: Actinomycetota) had a significantly high relative abundance in NJDZ, YNDZ, and WHDZ samples (Figure 1a and Figure 2a). However, Proteobacteria, Bacteroidetes, and Actinobacteria had a significantly higher relative abundance in WHDZ samples than in NJDZ and YNDZ samples. Among them, the maximum abundances of Proteobacteria, Bacteroidetes, and Actinobacteria were recorded in WHDZ01, NJDZ01, and WHDZ02, respectively (Figure 2c).

Proteobacteria is the most dominant seawater phylum [29]. Bacteroidetes can degrade some biological macromolecules such as chitin and cellulose [30]. Actinobacteria can degrade organic pollutants and play an important role in marine pollution remediation [31,32]. In addition, some phyla with a low abundance were also recorded, such as Cyanobacteria and Firmicutes, among which Cyanobacteria had a high abundance in YNDZ02 and YNDZ03, while Firmicutes had a high abundance in YNDZ01 and YNDZ02. Vieira et al. pointed out that Firmicutes may be related to sea pollution; therefore, sea areas with high Firmicutes abundance may indicate pollution [32]. Nonmetric multidimensional scaling showed that the microbial communities of the NJDZ and WHDZ samples had a similar species composition, and the microbial community of the YNDZ samples had a different species composition (Figure 1b). Environmental and ecological selection may play a role in generating and maintaining microbial diversity in a geographically defined and seemingly unstructured marine ecosystem. Algal microbiomes are complex and dynamic, and their diversity may be driven by ecological or environmental selection to generate and maintain these intimate relationships over space and evolutionary time [33,34].

The most abundant microbial class among the nine macroalgae samples also significantly differed according to region. The dominant macroalgae-associated bacteria of the NJDZ samples were mostly Gammaproteobacteria, Flavobacteriia, Alphaproteobacteria, and Saprospiria; the dominant macroalgae-associated bacteria of the YNDZ samples were mostly Alphaproteobacteria, Gammaproteobacteria, Actinobacteria, Flavobacteriia, Acidimicrobiia, and Bacilli; and the dominant macroalgae-associated bacteria of the WHDZ samples were mostly Alphaproteobacteria, Flavobacteriia, Gammaproteobacteria, Actinobacteria, and Acidimicrobiia (Table 1). It is worth noting that Gammaproteobacteria was the dominant class in the NJDZ samples, while Alphaproteobacteria was the dominant class in the WHDZ and YNDZ samples (Figure 2b,d). These results indicate that the community composition of the macroalgae-associated bacteria is not only related to the environment, but also affected by the host, in accordance with natural selection.

### 2.4. Functional Prediction of Bacterial Communities

DIAMOND software (Version 0.9.9.110) was used to annotate the database of common functions of nonredundant gene sets (e-value ≤ 10^−5^); there were 2,485,201 predicted genes after the original redundancy was removed, and 1,493,359 (60.09%) genes could be compared with the KEGG database, among which 806,765 (32.46%) genes could be compared with 7566 KEGG ortholog groups. There were 1,428,379 (57.48%) genes that could be compared with the eggNOG database, and 61,807 (2.49%) that could be compared with the CAZy database (Table 2).

In the functional level of the communities, there were significant differences in the composition and abundance of functional genes of macroalgae-associated bacteria distributed in different spaces on the surface of red algae. Regarding the community functional level, there were significant differences in the composition and abundance of functional genes of macroalgae-associated bacteria on the surface of red algae among regions. WHDZ samples had a higher genetic diversity regarding all of the functional properties, while YNDZ samples had a lower genetic diversity (Figure 3). The distribution of KEGG subclasses (level 1) showed more genes related to various metabolism pathways, followed by genetic information processing pathways (Figure 3a,b). The relative abundance of YNDZ01 enriched in various pathways was the lowest, and the relative abundance of YNDZ02 enriched in human disease pathways was the highest. These predicted genes were significantly enriched in amino acid metabolism, carbohydrate metabolism, energy metabolism, metabolism of cofactors and vitamins, membrane transport, nucleotide metabolism, and cellular community-prokaryotes. A principal components analysis plot showed that the microbial communities in NJDZ, YNDZ, and WHDZ samples differed in terms of functional structure (Figure 3c).

The annotated genes in the Clusters of Orthologous Genes database are divided into 21 Clusters of Orthologous Genes functional classes. In this analysis, a large number of contigs were classified as “function unknown (S)”, “amino acid transport and metabolism (E)”, “energy production and conversion (C)”, “replication, recombination and repair (L)”, “cell wall/membrane/envelope biogenesis (M)”, and “inorganic ion transport and metabolism (P)” (Appendix A).

### 2.5. CAZymes Insights on the Bacterial Community

The BLASTX results of the CAZy database are combined with the hierarchical structure of the database to obtain each level of CAZymes information. Comparing the various strains of the six families with the CAZy database (Figure 4a) revealed 2,151,233 carbohydrate active enzyme genes, among which the GH family had the most genes (24,756), followed by the family of glycosyltransferase (GT) genes (20,694), carbohydrate-binding module (CBM) family genes (11,291), carbohydrate esterase family genes (3079), PL family genes (2564), and auxiliary activity family genes (1539). The strains with more GH family genes also had more GT and CBM family genes (Figure 4b), which might be related to the functional correlation of enzymes in each family. The most abundant families predicted in this genome were GH28, GH38, GT2, GT4, CBM13, GT51, GH3, CBM6, CBM50, and GH23 from level 2 of the CAZymes database (Figure 4c). Further analysis showed that the strains with a high carbohydrate active enzyme gene richness were mainly from the Pseudoalteromonadaceae of Proteobacteria and Flavobacteriaceae of Bacteroidetes.

Marine algae are the most promising raw material for the replacement of land plants, and algal oligosaccharides are degraded and utilized by marine heterotrophic microorganisms [35]. Many enzymes in algae-related microorganisms, such as GH and PL, are involved in this important metabolic process [6,35]. The GH family can hydrolyze the glycosidic bonds between two or more carbohydrates or between carbohydrate and noncarbohydrate components, and are essential in the red algae polysaccharide hydrolyzing process [36]. The GT family is involved in the biosynthesis of disaccharides, oligosaccharides, and polysaccharides by catalyzing the transfer of glycosylates from activated donor molecules to specific receptors to form glycosidic bonds. The most widely distributed GT2 family proteins have the activities of cellulose synthetase, chitin synthetase, hyaluronic acid synthetase, glucan synthetase, and mannan synthetase [37]. These enzymes are related to the synthesis of algae polysaccharides, indicating that these macroalgae-associated bacteria can not only degrade algae polysaccharides, but also synthesize related polysaccharides themselves.

The main components of red algae are agar and carrageen. Research on carrageenase is the earliest and most extensive; κ-carrageenase belongs to the GH16 family and τ-carrageenase belongs to the GH82 family. β-Agarase belongs to the GH16, GH39, GH50, GH86, and GH118 families, while α-agarase belongs to the GH96 and GH117 families [38]. β-agarase in the GH16 and GH86 families is an endonuclease, β-agarase in the GH50 family has exonuclease activity, and α-agarase in the GH117 family is an exonuclease [39]. Therefore, when macroalgae-associated bacteria communities contain a variety of GH family enzyme coding genes, there is an opportunity for the emergence of endo-cut β-agarase, exo-cut β-agarase, and exo-cut α-agarase systems, which eventually degrade polysaccharides into oligosaccharides.

Annotation of the functional genes of the macroalgae-associated bacteria genome revealed that the bacteria genome isolated from the surface of red algae contained a large number of genes related to the degradation of alginate. This indicated that most of the strains on the surface of the algae had the basic function of degrading alginate, which was related to their living environment and their own carbon source utilization function [40,41]. Therefore, the diversity of these functional genes was further analyzed, revealing the presence of the PL7 family in strains distributed in Proteobacteria and Bacteroidetes. These results indicate that the microorganisms on the surface of algae are rich in alginate lyase, and that algae degradation requires the coordination of enzymes with multiple functions. These enzymes work together to utilize algae polysaccharides.

### 2.6. Mining of Enzymes for Algal Polysaccharide Degradation

Increasing attention has been paid to marine oligosaccharides due to their bioactivity, solubility, and bioavailability. Studies have shown that algae oligosaccharides have many potential applications [42]. The in-depth functional analysis of nine macroalgae metagenomic data was performed to identify new algal-polysaccharide-degrading enzymes for biotechnological purposes (Table 3 and Figure 5). Metagenomic data were then further analyzed to explore the number of algal-polysaccharide-degrading enzyme genes. There were significant differences in algal-polysaccharide-degrading enzymes on the surface of red algae among the remote regions. The number of alginate lyase, agarose, and carrageenase genes on the surface of red algae among the regions is shown in Table 3. The number of agarase genes from WHDZ was 1076, and that from NJDZ was 939, while YNDZ had only 41. The number of carrageenase genes from WHDZ was 940, and that from NJDZ was 759, with YNDZ only having 33. The number of alginate lyase genes from WHDZ was 840, and that from NJDZ was 829, while YNDZ had only 1. The agarase gene was mainly distributed in WHDZ03, followed by NJDZ01. There were very few agarase genes in YNDZ, with only 3 alginate lyase genes in YNDZ01 and 38 in YNDZ03. The carrageenase gene was mainly distributed in WHDZ03, followed by NJDZ01. Similarly, there were very few carrageenase genes in YNDZ, with only 3 carrageenase lyase genes in YNDZ01 and 30 in YNDZ03. The alginate lyase gene was mainly distributed in NJDZ01, followed by WHDZ03. There were also very few alginate lyase genes in YNDZ, with only one in YNDZ03.

These results suggest that WHDZ and NJDZ had a high abundance of algal-polysaccharide-degrading enzymes, while YNDZ had an extremely low abundance. There is evidence that the characteristics of the algae itself, secretions at different growth stages (small organic molecules such as early amino acids and organic acids, and large organic molecules such as algal polysaccharides and lipids), and environmental conditions for algal growth (such as pH, water flow, light, temperature, and nutrients) affect the composition of the macroalgae-associated bacterial community, and thus enzyme abundance [43,44,45,46]. Of note, there were significant differences in the algal-polysaccharide-degrading enzymes on the surface of red algae among the remote regions, especially in the NJDZ samples. The environmental differences between NJDZ, WHDZ, and YNDZ may impose a strong geographic differentiation in the biodiversity of algal microbiomes and their expressed enzyme genes.

Numerous macroalgae-associated marine microorganisms contain undeveloped enzymes such as GH and PL, which are involved in important metabolic pathways and whose characterization has the potential to provide new insights into the marine carbon cycle [24]. This study effectively extends our understanding of the diversity of algae surface microorganisms and their polysaccharide-degrading enzymes. It also deepens our understanding of how microorganisms process the hundreds of millions of tons of polysaccharides produced by algae each year and their role in the marine carbon cycling system.

### 2.7. Environmental Factor Analysis

Canonical correspondence analysis (CCA) and redundancy analysis (RDA) were performed using R software (Version 2.15.3), and relationships between bacterial communities and environmental variables were constructed at the genus level. As shown in Figure 6a,b, the first and second spindles of the CCA accounted for 56.85% and 43.15% of the variance in the relative abundance of the bacterial community, respectively, and the first and second spindles of RDA accounted for 53.46% and 46.54%, respectively. This indicated that the relationship between the bacterial community and environmental variables was reliable. The CCA and RDA analyses of environmental factors showed that, without considering the species and composition of algae, pH and temperature were the main environmental factors affecting bacterial community structure. Most previous studies have compared native species diversity (alpha diversity) at different latitudes [46]. However, recent studies have shown that temperature is also one of the most important variables explaining the differences in local community species composition over large-scale latitudinal gradients [34,47].

Salinity, temperature, pH, and DO are the main environmental factors affecting macroalgae-associated bacteria communities. They not only affect the community structure and distribution of bacteria, but also can be used to indicate the environmental status of marine ecosystems [44,45]. In order to further explore the influence of environmental factors on the macroalgae-associated bacteria communities, a correlation heat map analysis was conducted, and the results are shown in Figure 7. *Limnothrix* and *Leptolyngbya* were positively correlated with temperature and pH, but negatively correlated with DO. *Psychroflexus* and *Psychromonas* were positively correlated with DO, but negatively correlated with temperature and pH. In addition, *Maribacter*, *Nitratireductors*, and *Robiginitomaculum* were negatively correlated with salinity, and *Acinetobacter* was positively correlated with salinity. This showed that environmental factors had varying effects on different macroalgae-associated bacteria.

### 2.8. Diverse Bacterial Lineages Potentially Harbor Antibiotic Resistance Genes

This study further screened the diversity and abundance of the antibiotic resistance genes (ARGs) in the microbial gene catalog. The ARG-like genes were assigned to 20 ARG classes, with high proportions being unclassified (Appendix A). The total numbers of ARG classes in YNZD02 were surprisingly higher than in other samples, suggesting that human activities may also lead to ARG enrichment [48,49]. YNZD had a higher relative abundance of ARG classes compared with NJDZ and YNDZ (Appendix A). These findings suggest that macroalgae are a neglected but potentially important reservoir of ARGs. The main ARG classes detected in YNZD02 were *TEM-171* and *tetG* (tetracycline resistance) genes (Appendix A). These findings suggest that macroalgae in Halmahera, Indonesia are possibly contaminated by anthropogenic ARGs to a certain extent. The diversity of ARG hosts with multiple antibiotic potential resistances suggests that these bacteria are ARG hosts and may play a key role in the acquisition and spread of antibiotic resistance in macroalgae.

ARGs are present in almost all environments. They are either endemic to the natural environment or derived from human-dominated ecosystems [50,51]. However, the diversity and hosts of ARGs in macroalgae remain unclear. Their abundance of microbes indicates that macroalgae could be a potential reservoir for ARGs. If these microbes that are resistant to various antibiotics are algal pathogens, this could worsen an outbreak of macroalgal disease. In addition, macroalgal ecosystems may be contaminated by antibiotics and ARGs from human and agricultural wastes [52]. Residues of antibiotics in aquatic systems can be deposited in macroalgae, and may eventually threaten their growth and affect microbial ecology [53]. Therefore, studying ARGs is an important component of evaluating macroalgae ecosystem health.

## 3. Materials and Methods

### 3.1. Macroalgae Sampling and Characterization

Three macroalgae samples (NJDZ01, NJDZ02, and NJDZ03) were collected from the Great Wall Station in Antarctica (lat: 62°12′, long: 58°57′) during China’s 34th Antarctic expedition, another three (YNDZ01, YNDZ02, and YNDZ03) from Halmahera in Indonesia (lat: 0°36′, long: 127°52′), and a final three (WHDZ01, WHDZ02, and WHDZ03) from Weihai in China (lat: 37°25′, long: 122°17′). The nine macroalgae samples, identified as *Palmaria decipiens* (NJDZ01), *Curdiea racovitzae* (NJDZ02), *Iridaea cordata* (NJDZ03), *Laurencia japonica* (YNDZ01), *Amphiroa foliacea* (YNDZ02), *Gracilaria tenuistipitata* (YNDZ03), *Grateloupia filicina* (WHDZ01), *Chondrus ocellatus* (WHDZ02), and *Hyalosiphonia caespitosa* (WHDZ03), were collected from the seashore. The algal samples were washed with sterile seawater and brought back to the laboratory for collection of surface-associated bacteria.

Samples were then washed again with sterile seawater in the laboratory. After washing, sterile scissors were used to cut the algae into smaller chunks. They were then placed in a sterile tube with an appropriate amount of sterilized seawater and vortexed three times for 40 s each. The liquid was filtered onto 0.22 μm filter membranes (Merck Millipore, Darmstadt, Germany) to collect the associated bacteria.

### 3.2. DNA Extraction and Sequencing

The FastDNA Spin Kit for Soil (MP Bio, Santa Ana, CA, USA) was used to extract genomic DNA from the macroalgae-associated microorganisms. After genomic DNA extraction was completed, the extracted genomic DNA purity and integrity were analyzed using agarose gel electrophoresis (Appendix A). To conduct metagenomic sequencing of the DNA samples of the macroalgae-associated microorganisms, the qualified genomic DNA was sent to Novogene Bioinformatics Technology Co., Ltd. (Beijing, China) for sequencing, assembly, and functional annotation.

After base calling, the result file stored in FASTQ format, called a raw read, was obtained. Since the original dismounting data may contain data with joints (introduced during database construction) and a certain proportion of low-quality data (generated with sequencing reading), in order to ensure the accuracy and reliability of subsequent analysis results, Trim Galore was used to preprocess raw data and obtain clean data for subsequent analysis. In the preprocessing, the junction sequence in sequencing reads, sequences with average mass value <25, truncated sequences with a length <70 bp, and host sequence contaminations were removed. Due to the impact of the high-throughput sequencing error rate on the results, it was necessary to evaluate the quality of the optimized data.

### 3.3. Metagenome Assembly

After pretreatment, clean data were obtained, and assembly analysis was performed using MEGAHIT assembly software (Version 1.0.4, http://www.l3-bioinfo.com/products/megahit.html accessed on 8 December 2022) [54]. The assembled scaffolds were broken at the N joint to produce N-free sequence fragments called scaftigs. Clean data of samples after quality control were compared with scaftigs after sample assembly using Bowtie2 software (https://bowtie-bio.sourceforge.net/bowtie2/index.shtml accessed on 10 December 2022) [55], and unused pair-end reads were obtained. The unused reads of each sample were combined together for mixed assembly with the same assembly parameters as those of a single sample. Then the assembled scaftigs were interrupted from N connection and the scaftigs without N were left. For scaftigs generated by single-sample and mixed assembly, fragments <500 bp were filtered out, and statistical analysis and subsequent gene prediction were performed.

### 3.4. Gene Prediction and Abundance Analysis

MetaGeneMark (Version 2.1, http://topaz.gatech.edu/GeneMark/meta_gmhmmp.cgi accessed on 11 December 2022) [56] was utilized to predict the open reading frame (ORF) of each sample and mixed assembled scaftig (≥500 bp). Based on the predicted result, information <100 nt was filtered out. For ORF prediction results of each sample and mixed assembly, CD-HIT software (Version 4.5.8, http://www.bioinformatics.org/cd-hit/ accessed on 12 December 2022) [57] was used to remove redundancy in order to obtain a nonredundant initial gene catalogue. By default, a 95% identify and 90% coverage were used for clustering, and the longest sequence was selected as the representative sequence. Clean data of each sample were compared with the initial gene catalogue with Bowtie2, and the number of gene reads in each sample was calculated. The number of reads ≤2 in each sample was filtered out, and the final gene catalogue (unigenes) for subsequent analysis was obtained. Based on the number of reads and gene length, the abundance of each gene in each sample was calculated.

### 3.5. Taxonomy Prediction

DIAMOND software (Version 0.9.9, https://github.com/bbuchfink/diamond/ accessed on 15 December 2022) [58] compared unigenes with bacteria, fungi, archaea, and viruses extracted from the National Center for Biotechnology Information nonredundant (NR) database (Version 2018.01). For the comparison results of each sequence, e values ≤ the smallest e value* 10 were selected for subsequent analysis. After filtering, since each sequence may have multiple comparison results, multiple different species’ classification information could be obtained. In order to ensure its biological significance, the lowest common ancestor algorithm (systematic classification applied to MEGAN software Version 4) was adopted to apply the classification level before the first branch as the species annotation information of the sequence. Based on the results of lowest common ancestor annotation and gene abundance table, the abundance of each sample in each taxonomic level (kingdom, phylum, class, order, family, genus, and species) was obtained.

### 3.6. Functional Annotations

DIAMOND software (Version 0.9.9, https://github.com/bbuchfink/diamond/ accessed on 16 December 2022) [58] was used to compare unigenes with various functional databases. For the comparison results of each sequence, the results with the highest score were selected for subsequent analysis. Based on the comparison results, the relative abundance of different functional levels was calculated (the relative abundance of each functional level was equal to the sum of the relative abundances of the genes annotated as this functional level). The Kyoto Encyclopedia of Genes and Genomes (KEGG) database (Version 2018-01-01, http://www.kegg.jp/kegg/ accessed on 18 December 2022) [59] was divided into six levels, the eggNOG database (Version 4.5, http://eggnogdb.embl.de/#/app/home accessed on 18 December 2022) [60] into three levels, and the CAZy database (Version 201801, http://www.cazy.org/ accessed on 18 December 2022) [61] into three levels. Based on the results of functional annotation and gene abundance table, the gene number table of each sample at each classification level was obtained.

## 4. Conclusions

Among the abundance of living organisms in the ocean, algae are the main source of carbon and carry out photosynthesis for microorganisms. Marine algae contain rich polysaccharide substances, and their polysaccharide degradation products are widely used in medicine, food, and cosmetics industries. Therefore, it is very important to understand the degradation strains and enzymes of algae-related microbes. The results of this study showed that the microbial community composition was closely related to the living environment of the algae, with obvious regional differences, and was greatly affected by temperature and DO. By analyzing the genomic data of isolated strains on red algae surfaces, the diversity of related bacteria and polysaccharide-degrading enzyme systems was discussed. These findings broaden the research into marine polysaccharide-degrading enzymes, and provide reference for the subsequent development of macroalgae polysaccharide-degrading bacteria and enzymes.

## Figures and Tables

**Figure 1 ijms-24-11019-f001:**
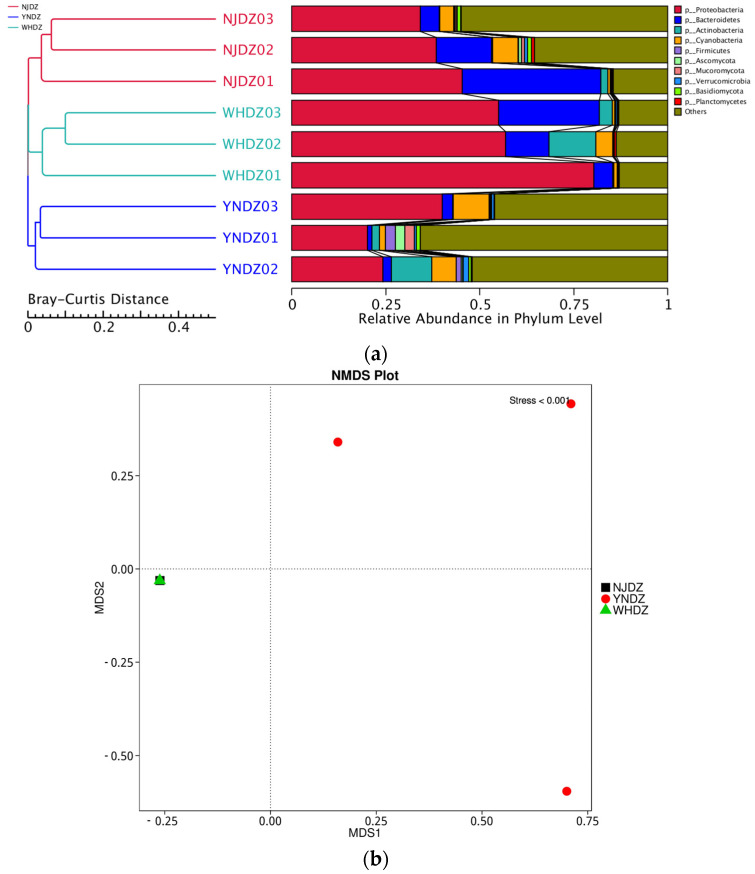
Relative abundance of species at phylum level. (**a**) Sample clustering analysis; on the left is Bray–Curtis distance clustering tree structure; on the right is the relative abundance distribution of each sample at phylum level. (**b**) NMDS analysis; each point in the figure represents a sample, the distance between points represents the degree of difference, and samples in the same group are represented by the same color; the stress is less than 0.2, indicating that the NMDS analysis has a certain reliability.

**Figure 2 ijms-24-11019-f002:**
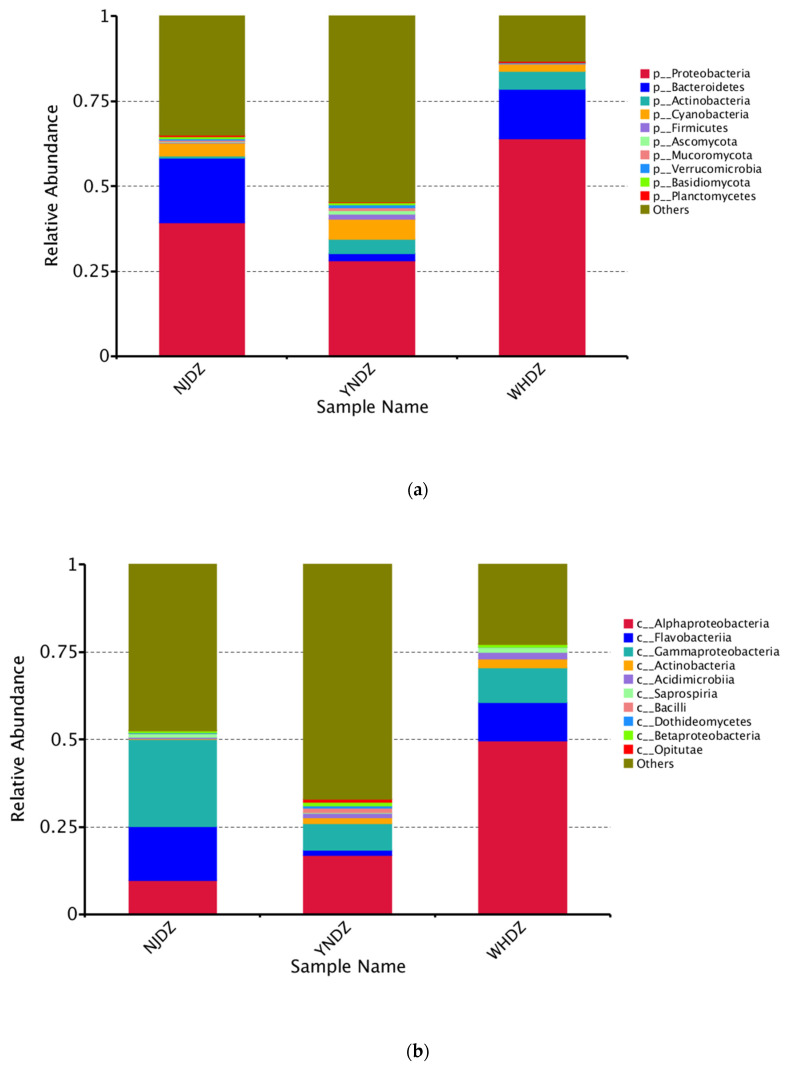
Relative species abundance at phylum and class levels. (**a**,**c**): Relative species abundance at phylum levels. (**b**,**d**): Relative species abundance at class levels.

**Figure 3 ijms-24-11019-f003:**
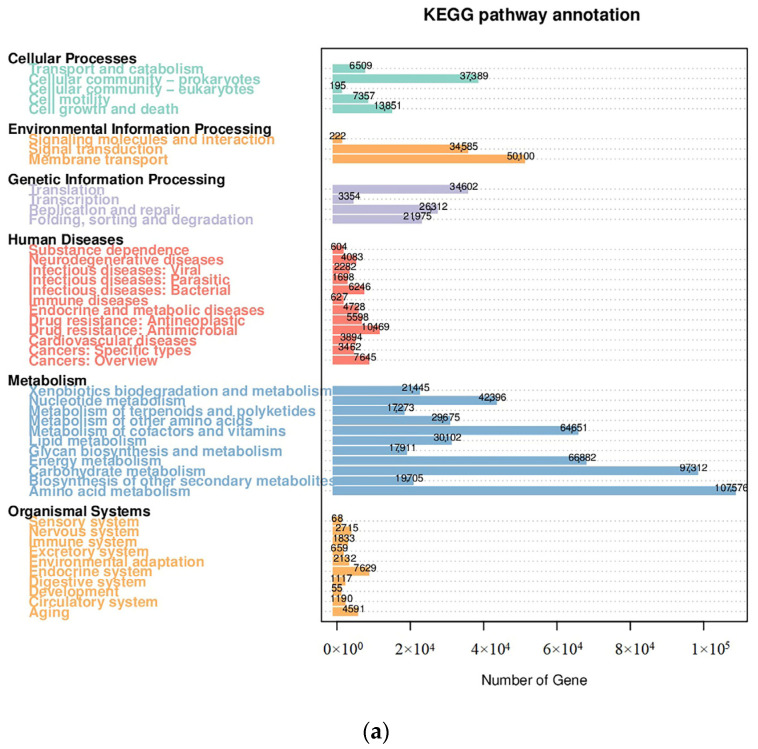
Functional prediction of bacterial communities. (**a**): Distribution of KEGG subclasses; the numbers on the bar chart represent the number of unigenes on the note. (**b**): KEGG database relative abundance column, with the vertical axis showing the relative proportion of comments to a feature class. (**c**): PCA analysis of KEGG functional abundance; each point in the diagram represents a sample, and samples from the same group are represented using the same color.

**Figure 4 ijms-24-11019-f004:**
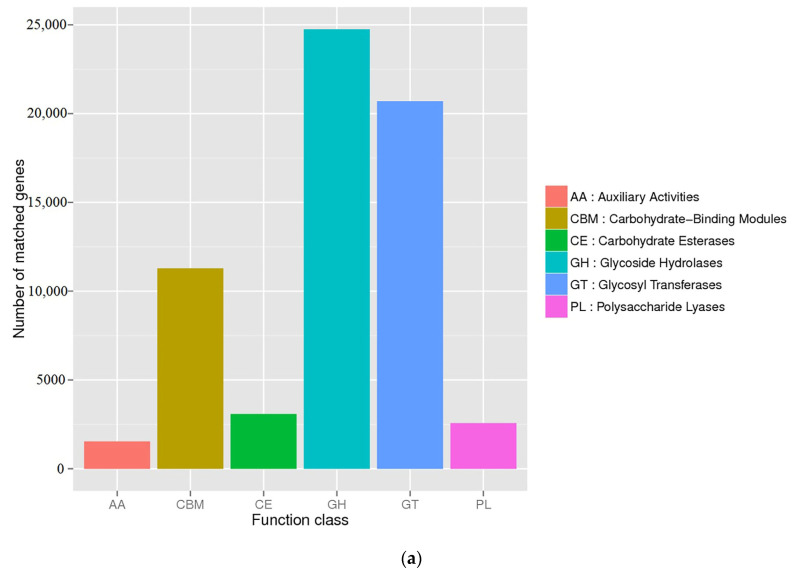
Annotation of CAZy enzyme. (**a**) Notes on CAZy unigenes. (**b**) CAZy level 1 relative abundance. Sample clustering analysis; on the left is Bray–Curtis distance clustering tree structure; on the right is the relative abundance distribution of each sample. (**c**) CAZy level 2 relative abundance.

**Figure 5 ijms-24-11019-f005:**
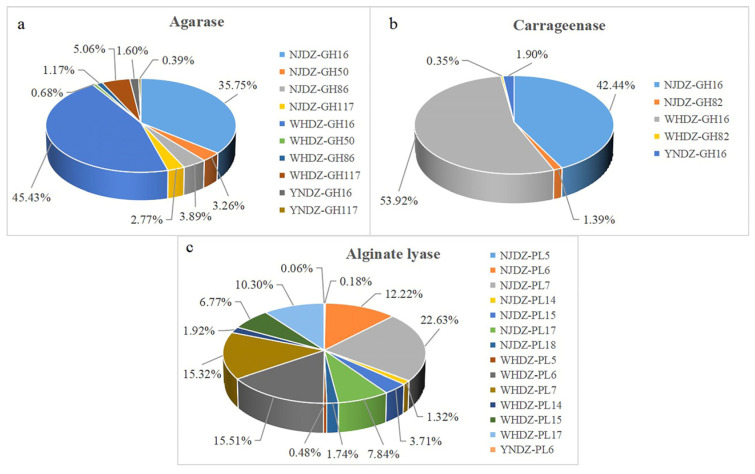
Relative abundances of agarase (**a**), carrageenase (**b**), and alginate lyase (**c**) in CAZy family from the metagenomic sample.

**Figure 6 ijms-24-11019-f006:**
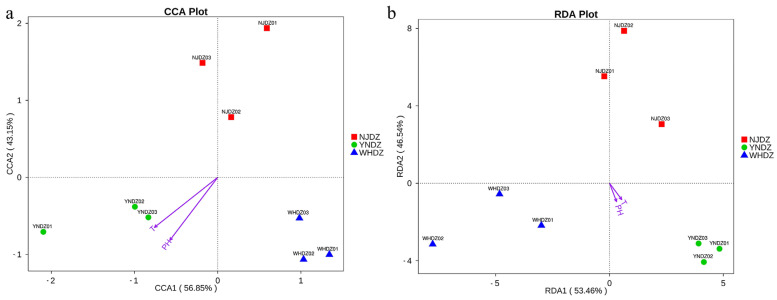
CCA (**a**) and RDA (**b**) of bacterial and environmental factors at the genus level. The points in the figure represent the samples, the environmental factors are represented by arrows, and the length of the arrows indicates the degree of correlation (the longer the arrows, the greater the correlation).

**Figure 7 ijms-24-11019-f007:**
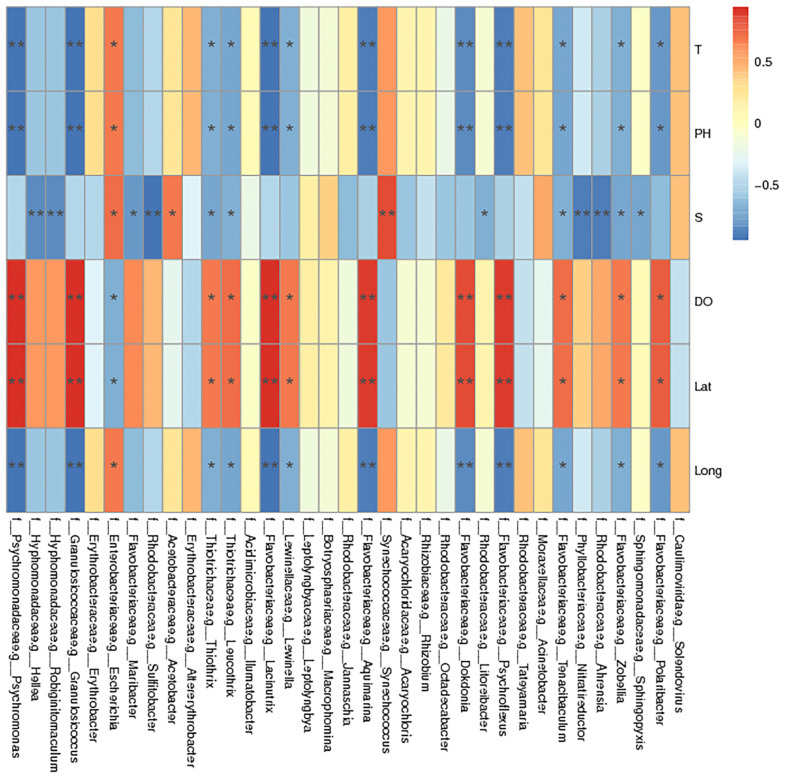
Correlation analysis between macroalgae-associated bacteria and environmental factors. The horizontal axis is the species information, the vertical axis is the environmental factor, the R-value is displayed in different colors in the figure, and the legend on the right side is different color intervals; * 0.01 ≤ *p* ≤ 0.05, ** 0.001 ≤ *p* ≤ 0.01.

**Table 1 ijms-24-11019-t001:** The physiochemical parameters of overlying seawater among three remote spaces and the corresponding most abundant associated bacteria genera.

Location	Lat	Long	DO	pH	Salinity	Temperature	The Dominant Macroalgae-Associated Bacteria
Great Wall Station	62.2°	58.95°	10.836 mg/L	8.01	34.07‰	−1.34 °C	Gammaproteobacteria, Flavobacteriia, Alphaproteobacteria, Saprospiria
Halmahera	0.6°	127.86°	5.44 mg/L	8.79	34.7‰	27.9 °C	Alphaproteobacteria, Gammaproteobacteria, Actinobacteria, Flavobacteriia
Weihai	37.42°	122.28°	6.65 mg/L	8.43	32.4‰	8 °C	Alphaproteobacteria, Flavobacteriia, Gammaproteobacteria, Actinobacteria

**Table 2 ijms-24-11019-t002:** The functional prediction of bacterial communities.

Gene catalogue	2,485,201
Annotated on KEGG	1,493,359 (60.09%)
Annotated on KO	806,765 (32.46%)
Annotated on KO number	7566
Annotated on EC	493,520 (19.86%)/2517
Annotated on pathway	507,914 (20.44%)/409
Annotated on eggNOG	1,428,379 (57.48%)
Annotated on OG	1,428,379 (57.48%)/31,380
Annotated on CAZy	61,807 (2.49%)

**Table 3 ijms-24-11019-t003:** The predicted proteins containing CAZymes modules acting on algal polysaccharide.

Location	Enzyme Number	E.C. Number	CAZy Family
Agarase			
NJDZ	939	EC 3.2.1.81	GH16
		EC 3.2.1.81	GH50
		EC 3.2.1.81	GH86
		EC 3.2.1.-	GH117
WHDZ	1076	EC 3.2.1.81	GH16
		EC 3.2.1.81	GH50
		EC 3.2.1.81	GH86
		EC 3.2.1.-	GH117
YNDZ	41	EC 3.2.1.81	GH16
		EC 3.2.1.-	GH117
Carrageenase			
NJDZ	759	EC 3.2.1.83	GH16
		EC 3.2.1.157	GH82
WHDZ	940	EC 3.2.1.83	GH16
		EC 3.2.1.157	GH82
YNDZ	33	EC 3.2.1.83	GH16
Alginate lyase			
NJDZ	829	EC 4.2.2.3	PL5
		EC 4.2.2.3	PL6
		EC 4.2.2.-	PL7
		EC 4.2.2.3	PL14
		EC 4.2.2.3	PL15
		EC 4.2.2.3	PL17
		EC 4.2.2.3	PL18
WHDZ	840	EC 4.2.2.3	PL5
		EC 4.2.2.3	PL6
		EC 4.2.2.-	PL7
		EC 4.2.2.3	PL14
		EC 4.2.2.3	PL15
		EC 4.2.2.3	PL17
YNDZ	1	EC 4.2.2.3	PL6

## Data Availability

The dataset of the metagenome supporting the conclusions of this article is available in the NCBI repository: http://www.ncbi.nlm.nih.gov/bioproject/974727 accessed on 21 May 2023 (BioProject ID PRJNA974727).

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
