# Peer review of "Metagenomic Insights Reveal the Microbial Diversity and Associated Algal-Polysaccharide-Degrading Enzymes on the Surface of Red Algae among Remote Regions"

_ijms, 2023, doi:10.3390/ijms241311019_

Round 1

Reviewer 1 Report

The manscript show a big investment in terms of experimentation and data production. It is interesting and could contribute to the knoledge in the field. It will surely appretiated by the commmunity. However to be published it needs to be refined in some parts. Following are my suggestions:

Notes:

- Reference number 3 and 5 cited at line 35 are the same;

- check all the reference, i.e, ref numebr 6 : there is a "-" in the middle of the word "hydrolysates" and the woed "anti-inflammatory" has an extra "
fl"; ref number 35: each word has a capital letter and the species name in not in italics.

Introduction

Following the Instructions for authors, I think the underlined point have not been clearly addressed in deep. Every aspect of the manuscript aims background have been discussed jus on surface and also waht are the main aims and the main results are not described well. In addition, I think a major enphasis on the metagenomics should be done together with the biodiversity offered byt hte different geographic niches. Pkease implement the introduction.

"The introduction should briefly place the study in a broad context and highlight why it is important. It should define the purpose of the work and its significance, including specific hypotheses being tested. The current state of the research field should be reviewed carefully and key publications cited. Please highlight controversial and diverging hypotheses when necessary. Finally, briefly mention the main aim of the work and highlight the main conclusions. Keep the introduction comprehensible to scientists working outside the topic of the paper."

Material and Methods

-line 101, "(Gui et al., 2019)": is it a citation? if yes dollow the rule for citation

-line 105, "Coralline algae (YNDZ02), Porphyra (YNDZ03)": which species?

-line 118: please add in supplementary materials the DNA quality control gel. In addition, "Qualified DNA samples were detected", what do you mean?

-line 122: "The original data obtained by sequencing was the image file.", what it means?

-line 130-131: "Due to the impact of the high-throughput sequencing error rate on the results, it was necessary to evaluate the quality of the optimized data", how you did it? where are the data?

-Paragraph 2.3, 2.4, 2.5, 2.6:

-please clarify the scaftigs generation; you speak of N-free sequence fragments, but in the end the fragments are just three for each sample, is it right?

-Please add the citation for each software utilized.

Results and Discussion

-line 206: what do you means for "boundary level"?

-246-257: please add a table resuming the characteristic of the sampling sites and the corresponding most abundant associated bacteria genera;

-line 260: "predicition" = predicted genes

-lines 259-264: make a table of this results;

-lines 266-268: review the english;

line 267: "macroalgae-associated bacteria on the surface of", you mean the strictly associated bacteria?

-lines 270-271: "greater number of genes related to various metabolism pathways", this expression is not professional nor scientific. What does it mean? the results description should precise and strict;

-line 272: "Macroalgae-associated bacteria communities" which one? You already speak about functional class associated...what is now the point? I think you should make this paragraph more detailed and clear remarking the right differences among sites, macroalgae species, bacteria type of association etc.;

-Figure 3 Caption : please add details about the general topic of the figure and add detail for each panel;

-Lines 287-289: please reformlate this sentence, too many repetion of the word "Cazy" make it roundabout;

-Figure 4: add detailend caption. In the text explain it better, particularly the panel b and the phylogenetic relevance and connection in respct to the functional baundances;

-line 310: Ref 36 is wrong. the argument of the cited reference is not related to the GH families;

-Figure 6: add more details to the caption;

-Figure 7: again add more details to the caption for example, what doesit means the asterisks?

- I would avoid too much speculation of the results based on bibliography. Manage the discussion of the results making less sure assumption .

Author Response

Response to Reviewer 1 Comments

Dear editor and reviewer:

Thank you for reviewing our manuscript titled “Metagenomic insights reveal the microbial diversity and associated algal polysaccharide-degrading enzymes on the surface of red algae among remote regions”. Those comments are all valuable and very helpful for revising and improving our manuscript, as well as the important guiding significance to our researches. We have studied comments carefully and have made correction which we hope meet with approval. Our point-by-point responses to the reviewers' comments are given below:

Point 1: Reference number 3 and 5 cited at line 35 are the same. 

Response 1: We have revised the reference 5 to be correct.

Point 2: Check all the reference, i.e, ref numebr 6 : there is a "-" in the middle of the word "hydrolysates" and the woed "anti-inflammatory" has an extra "fl"; ref number 35: each word has a capital letter and the species name in not in italics.

Response 2: We have checked all the references and corrected the errors in writing.

Point 3: Introduction

Following the Instructions for authors, I think the underlined point have not been clearly addressed in deep. Every aspect of the manuscript aims background have been discussed jus on surface and also waht are the main aims and the main results are not described well. In addition, I think a major enphasis on the metagenomics should be done together with the biodiversity offered byt hte different geographic niches. Pkease implement the introduction.

"The introduction should briefly place the study in a broad context and highlight why it is important. It should define the purpose of the work and its significance, including specific hypotheses being tested. The current state of the research field should be reviewed carefully and key publications cited. Please highlight controversial and diverging hypotheses when necessary. Finally, briefly mention the main aim of the work and highlight the main conclusions. Keep the introduction comprehensible to scientists working outside the topic of the paper." 

Response 3: Thank you very much for your instructive comments, we have studied comments carefully and have made correction which we hope meet with approval.

Point 4: line 101, "(Gui et al., 2019)": is it a citation? if yes dollow the rule for citation.

Response 4: We have deleted "(Gui et al., 2019)".

Point 5: line 105, "Coralline algae (YNDZ02), Porphyra (YNDZ03)": which species? 

Response 5: We have supplemented the species of YNDZ02 (Amphiroa foliacea), and YNDZ03 (Gracilaria tenuistipitata).

Point 6: line 118: please add in supplementary materials the DNA quality control gel. In addition, "Qualified DNA samples were detected", what do you mean?

Response 6: We have added the DNA quality control gel in supplementary materials (Figure S1).

"Qualified DNA samples were detected" means “To conduct metagenomic sequencing of the DNA samples of the macroalgae-associated microorganisms, the qualified genomic DNA was sent to Novogene Bioinformatics Technology Co., Ltd (Beijing, China) for sequencing, assembly and functional annotation” and we have made appropriate changes in the text.

Point 7: line 122: "The original data obtained by sequencing was the image file.", what it means? 

Response 7: We have deleted "The original data obtained by sequencing was the image file."

Point 8: line 130-131: "Due to the impact of the high-throughput sequencing error rate on the results, it was necessary to evaluate the quality of the optimized data", how you did it? where are the data?

Response 8: We have discussed this in Section 3.2. Sequencing and Metagenomic Assembly: The Illumina HiSeq sequencing platform produced a total of 81,611.46 Mbp of raw data (average data volume was 9,067.94 Mbp). After quality control, 81,245.68 Mbp of clean data were obtained (average data volume was 9,027.30 Mbp), and the effective data rate of the quality control was 99.55% (Table S1).

Point 9: Paragraph 2.3, 2.4, 2.5, 2.6:

-please clarify the scaftigs generation; you speak of N-free sequence fragments, but in the end the fragments are just three for each sample, is it right?

Response 9: We have clarified the scaftigs generation in Paragraph 2.3.

Point 10: Please add the citation for each software utilized.

Response 10: We have added the citation for each software utilized in paragraph 2.3, 2.4, 2.5, and 2.6.

Point 11: line 206: what do you means for "boundary level"? 

Response 11: line 206: "boundary level" is wrong and we have made the correct changes in 3.3. Microbial Diversity and Community Composition.

Point 12: 246-257: please add a table resuming the characteristic of the sampling sites and the corresponding most abundant associated bacteria genera.

Response 12: We have added a table resuming the characteristic of the sampling sites and the corresponding most abundant associated bacteria genera.

Point 13: line 260: "predicition" = predicted genes.

Response 13: We have made the correct changes.

Point 14: lines 259-264: make a table of this results.

Response 14: We have added a table of this results.

Point 15: lines 266-268: review the english. 

Response 15: We have reviewed the english of lines 266-268.

Point 16: line 267: "macroalgae-associated bacteria on the surface of", you mean the strictly associated bacteria?

Response 16: The associated bacteria, in this work, refer to that strictly associated to surface of macroalgae but not the simple attached microorganisms. The following is the collection method of associated bacteria: The algae surface was washed with sterilized seawater two or three times, then cut into pieces and put it into sterile tubes. Sterile seawater was added and eddy oscillations were used to shock it three times for 2 minutes each. Microorganisms were then extracted and collected using 0.22-to sterile filter membranes (50 mm).

Point 17: lines 270-271: "greater number of genes related to various metabolism pathways", this expression is not professional nor scientific. What does it mean? the results description should precise and strict.

Response 17: We have revised lines 270-271: "greater number of genes related to various metabolism pathways" to a more professional and scientific expression.

Point 18: line 272: "Macroalgae-associated bacteria communities" which one? You already speak about functional class associated...what is now the point? I think you should make this paragraph more detailed and clear remarking the right differences among sites, macroalgae species, bacteria type of association etc.

Response 18: Thank you very much for your instructive comments, we have supplemented the content to make this paragraph more detailed and clear remarking the right differences among sites, macroalgae species.

Point 19: Figure 3 Caption : please add details about the general topic of the figure and add detail for each panel. 

Response 19: We have added details about the general topic of the figure and add detail for each panel of Figure 3.

Point 20: Lines 287-289: please reformlate this sentence, too many repetion of the word "Cazy" make it roundabout.

Response 20: We have reformlated this sentence to make it not roundabout.

Point 21: Figure 4: add detailend caption. In the text explain it better, particularly the panel b and the phylogenetic relevance and connection in respct to the functional baundances.

Response 21: We have added more details to the caption of Figure 4 and explain it better in the text.

Point 22: line 310: Ref 36 is wrong. the argument of the cited reference is not related to the GH families.

Response 22: We have changed the Ref 36 with a correct one.

Point 23: Figure 6: add more details to the caption. 

Response 23: We have added more details to the caption of Figure 6.

Point 24: Figure 7: again add more details to the caption for example, what doesit means the asterisks?

Response 24: We have added more details to the caption of Figure 7 and noted the meaning of the asterisk.

We also revised other sections of this manuscript to improve the presentation of our results.

Once again, thank you very much for your comments and suggestions.

Reviewer 2 Report

This manuscript describes diversity of taxonomy and carbohydrate metabolism genes of red-algae associated bacteria collected from polar, tropical, and temperate regions using metagenomic approach.

The large amount of data and the detailed bioinformatics analyses showed the difference and features of the bacterial taxonomy and polysaccharide metabolism related in each region.

However, the overall content of the research is informatics than molecular sciences, and the results are related to microbiology and ecology. Therefore, I think that contents of this manuscript do not match the readers of this journal.

I recommend to resubmit the manuscript to journals more related to microbiology, enzymology, or bio-ecology.

Author Response

Response to Reviewer 2 Comments

Dear editor and reviewer:

Thank you for reviewing our manuscript titled “Metagenomic insights reveal the microbial diversity and associated algal polysaccharide-degrading enzymes on the surface of red algae among remote regions”.

Point 1: This manuscript describes diversity of taxonomy and carbohydrate metabolism genes of red-algae associated bacteria collected from polar, tropical, and temperate regions using metagenomic approach.

The large amount of data and the detailed bioinformatics analyses showed the difference and features of the bacterial taxonomy and polysaccharide metabolism related in each region.

However, the overall content of the research is informatics than molecular sciences, and the results are related to microbiology and ecology. Therefore, I think that contents of this manuscript do not match the readers of this journal.

I recommend to resubmit the manuscript to journals more related to microbiology, enzymology, or bio-ecology. 

Response 1: Macroalgae and macroalgae-associated bacteria together constitute the most efficient metabolic cycling system in the ocean. Their interaction, especially the responses of macroalgae-associated bacteria communities to algae in different geographical locations, is mostly unknown. Moreover, these macroalgae-associated microbes have been shown to produce algae-specific polysaccharide-degrading enzymes, and are therefore an important resource for the discovery of novel polysaccharide-degrading enzymes. At present, there are few algal polysaccharide-degrading enzymes that have application prospects and can be produced commercially. Our work will effectively extend our understanding of the diversity of algal surface microorganisms and their polysaccharide-degrading enzymes, and contributes to an in-depth study of their metabolic characteristics, ecological functions, and applications. Meanwhile, these  algal polysaccharide-degrading enzymes from special habitats with excellent properties have great significance for the development of biocatalysts for industrial applications.

We also revised other sections of this manuscript to improve the presentation of our results.

Once again, thank you very much for your comments and suggestions.

Reviewer 3 Report

This study focuses on the metagenomic analysis of red algae surface microbiomes and the search for enzymes that degrade algal polysaccharides. The samples analyzed were taken in three remote regions - China, Antarctica and Indonesia. Significant differences in microbial diversity and metabolic potential of bacteria associated with macroalgae were observed in different regions. The manuscript is well written, but some corrections are needed.

Correct phyla names should be given according to the latest requirements (https://doi.org/10.1099/ijsem.0.005056).

Figures 1-4 are too small, so they are hard to see. Their quality should be improved. Authors should scale up the figures.

Figure 7 legend should be more extended. Need to clarify the meanings of the asterisks.

Lines 64-65. "Attached bacteria" sounds a bit weird in this context.I suppose to change it to "associated bacteria".

 Minor editing of English language required

Author Response

Response to Reviewer 3 Comments

Dear editor and reviewer:

Thank you for reviewing our manuscript titled “Metagenomic insights reveal the microbial diversity and associated algal polysaccharide-degrading enzymes on the surface of red algae among remote regions”. Those comments are all valuable and very helpful for revising and improving our manuscript, as well as the important guiding significance to our researches. We have studied comments carefully and have made correction which we hope meet with approval. Our point-by-point responses to the reviewers' comments are given below: 

Point 1: Correct phyla names should be given according to the latest requirements (https://doi.org/10.1099/ijsem.0.005056). 

Response 1: Thank you very much for your timely reminder. Actually, the names of forty-two phyla of prokaryotes has been validly published at October 2021, but our work have been finished before this time. According to your suggestion, we have updated the validly published name of prokaryotes at results part.

Point 2: Figures 1-4 are too small, so they are hard to see. Their quality should be improved. Authors should scale up the figures. 

Response 2: We have scaled up the figures and improved their quality.

Point 3: Figure 7 legend should be more extended. Need to clarify the meanings of the asterisks. 

Response 3: We have added more details to the caption of Figure 7 and noted the meaning of the asterisk.

Point 4: Lines 64-65. "Attached bacteria" sounds a bit weird in this context.I suppose to change it to "associated bacteria". 

Response 4: We have changed the "Attached bacteria" to "associated bacteria".

We also revised other sections of this manuscript to improve the presentation of our results.

Once again, thank you very much for your comments and suggestions.
